# Using a Virtual Avatar Teaching Simulation and an Evidence-Based Teacher Observation Tool: A Synergistic Combination for Teacher Preparation

**Craig Berg** [1,*] **, Lisa Dieker** [2] **and Raymond Scolavino** [1]

1    Teaching and Learning, School of Education, University of Wisconsin-Milwaukee, Milwaukee, WI 53210, USA; ni3@uwm.edu
2    Special Education, School of Education and Human Sciences, University of Kansas, Lawrence, KS 66045, USA; lisa.dieker@ku.edu
*    Correspondence: caberg@uwm.edu

**Abstract:** The authors describe the combination of a mixed-reality simulation program and a teacher observation software tool to gather and analyze both comments and data in a teacher preparation program. Current research on this simulation experience for teachers to practice their craft, called TLE TeachLivE, is presented, along with the general and specific uses of this mixed-reality simulator. The authors also describe a recently developed teacher observation web-based app called SeeMeTeach that provides a platform for evidence-based teacher observations both within the simulator and in real classroom settings. The authors provide a description of how the pair of tools work in concert to identify strengths and weaknesses of teacher–student discourse, student engagement in lessons, and classroom management. The synergetic use of these tools provides a low-risk opportunity to practice teaching while maximizing data gathering for optimizing feedback and coaching based on evidence. In merging TLE TeachLive and SeeMeTeach, our work examined the following research questions using a mixed-methods research design: (1) How can the teacher observation tool aid teacher educators in identifying and collecting data during a teacher observation regarding key and discrete factors in teacher–student interactions and student engagement when attempting to improve teaching effectiveness? And (2) Does the TLE TeachLivE simulation produce a realism that offers potential for a wide enough variation in the display of teaching skills so that teaching fingerprints emerge?

**Keywords:** teacher preparation; teaching simulation; teacher observation; teacher feedback; coaching; observation data

## 1. Introduction

The authors describe the combination of a mixed-reality simulation program and a teacher observation software tool to gather and analyze both comments and data in a teacher preparation program. Current research on this simulation experience for teachers to practice their craft, called TLE TeachLivE, is presented, along with the general and specific uses of this mixed-reality simulator. The authors also describe how a recently developed teacher observation web-based app called SeeMeTeach provides the platform for evidence-based teacher observations within the simulator and in real classroom settings. The authors describe how the pair of tools work in concert to identify strengths and weaknesses of teacher–student discourse, student engagement in lessons, and classroom management. The synergetic use of these tools provides a low-risk opportunity to practice teaching while maximizing data gathering for optimizing feedback and coaching based on evidence. In merging TLE TeachLive and SeeMeTeach, our work examined the following research questions using a mixed-methods research design: (1) How can the teacher observation tool aid teacher educators in identifying and collecting data during a teacher observation

regarding key and discrete factors in teacher–student interactions and student engagement when attempting to improve teaching effectiveness? And (2) Does the TLE TeachLivE simulation produce a realism that offers potential for a wide enough variation in the display of teaching skills so that teaching fingerprints emerge?

## 2. Teacher Quality and Impact on the Learner

Teachers are crucial and central to structuring lessons and fostering and maintaining learning. Teacher decisions leading to action or non-action can significantly impact students in ways that are positively related to the targeted goals of instruction, or conversely, teacher decisions can reduce the impact on learners to below the intended and desired state. Aside from students showing up for class and the unique challenges among individual students, researchers note that teacher quality is the single most important school-based factor influencing student learning and academic achievement [1–6].

Kane and Staiger [7] note that teachers score lowest for teaching skills that are critical to teacher–student interactions or facilitating discussions and communications between teacher and student or among students. These potentially powerful, essential teacher skills for developing and facilitating student-engaging activities and maintaining robust learning environments are often lacking [8]. For example, students' passivity in classrooms is apparently an unchanging issue over time [9–11], an indicator that teachers either do not possess these student-engaging skills or choose not to put them into practice.

However, fostering student engagement in lessons is a teaching skill impacted by appropriate interventions [12] when novices are learning how to teach. Increasing and optimizing student engagement [9,13] can be achieved by choosing and utilizing teaching strategies designed for student engagement combined with synergistic teacher behaviors. Strategic and purposeful teacher decision-making can be incorporated into lessons by future or practicing teachers and significantly impact student engagement rather than leaving it up to chance [14].

The focus of the work described within this paper is based on the following premises: (1) teacher decisions and teacher actions are key to the success of a lesson, (2) student actions are key indicators of how the teacher is engaging students in the lesson, (3) simulation tools can represent reality for the generation of simple and typical teacher and student actions for practice, and (4) the teacher observation tool described can gather and analyze data relevant to the first three premises, allowing data embedded into the feedback loop to impact the professional skills of a future teacher positively.

The authors report this work by describing the mixed-reality virtual teaching simulation, its use, and why it is helpful for teacher preparation. Then, a description of the teacher observation tools is provided, followed by how integrating these systems using data and feedback can impact and shape teacher practice.

## 3. A Mixed-Reality Simulation and Teacher Preparation

Teacher preparation, practice, and improvement experiences usually occur in real classrooms, which offer varying levels of complexity and contain uncontrolled factors, potentially overwhelming pre-service teachers who often are uneasy with rudimentary teaching skills. Alternatively, practice occurs under more artificial conditions, such as micro-teaching [15], where the adult peers of the pre-service teacher attempt to play the role of elementary, middle, or high school students.

Teaching avatars in a mixed-reality simulation is a technology that was not available until a team of educators and technology experts crafted an almost magical present-day form, exciting many across the country. This technology allows a pre-service teacher to instruct avatar students in a virtual environment, under a controlled setting, with multiple complexity levels, working on a subset of teaching skills. At the same time, more difficult challenges are dampened or removed from the simulation so novice teachers have an acceptable level of challenge.

This simulation, TLE TeachLivE, developed by the University of Central Florida (UCF), is used by several institutions, including the authors'. The creators envisioned this mixed-reality virtual teaching simulation providing safe practice for both novice and expert teachers to develop or refine their teaching skills. During this simulation, a teacher interacts with avatar students (see Figure 1) by asking questions, answering questions, and generally interacting with a small group of five students in a classroom setting. The avatars can ask questions, answer questions and exhibit various misbehaviors. The simulator is only a tool, but the teacher educator is the one who drives the way it is used. In a recent publication by [16], they provide examples of use in STEM areas including, but not limited to, developing classroom management skills, developing or refining teacher–student interaction strategies, and working on skills within teacher discourse (questioning, wait-time, prompting), all aligned with the current high-leverage practices in education [17].

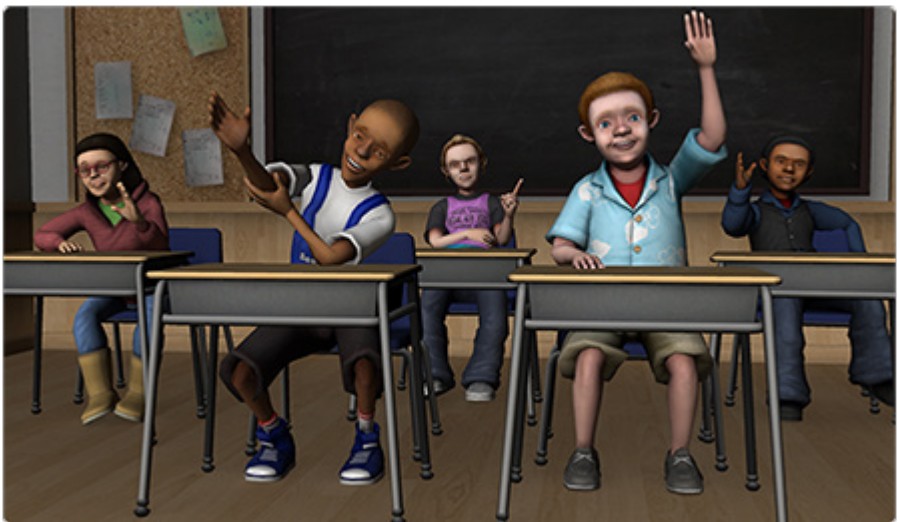

**Figure 1.** TLE TeachLivE middle school avatars. **Prior TLE TeachLivE research provides guidance and indicates positive effects.**

The avatar teaching environment provides low-risk ways for pre-service teachers to learn how to improve their teaching [16,18,19] in areas such as parent–teacher conferences [20]. They note that this type of environment is helpful for novices who still need to develop rudimentary skills. Others note it is quite useful for those who need to work on an array of skills, even for teachers in practice [16,21].

The research guiding the use of TLE TeachLivE indicates that the power of mixed-reality simulations comes from the strong foundation already established in the fields of medicine, military, and aviation [22]. In one study, the researcher found four 10 min sessions with the avatars can develop new teaching skills that carry over into real classroom instruction. This initial research was followed up with two additional studies, one with mathematics teachers and one with biology teachers, showing similar results [21,22]. These outcomes show that teacher educators now have a tool to assist in developing rudimentary skills with minimal risk. Despite limitations on the use of simulation (e.g., no group discussion, limited body movement of students) and the need to discuss potential bias in whatever make-up of the classroom one might select from the over 50 avatars now available for use (see Figure 2), the developers describe this tool as a safe environment to practice. Much like a pilot might practice flying in a simulator or a student might use one for preparing to drive, the focus in any educational simulation is not replacing "real" practice but to create automaticity in teacher behavior and prompt thinking about key skills the field has identified as being essential in teaching, like the HLPS or the Next-Generation Science Standards [23]. The point of any simulation, whether it be a card game, a case study, a standardized patient model, or an online game (e.g., Sim School), is to practice under the watchful eye of an expert reviewing the accuracy of the practice [16].

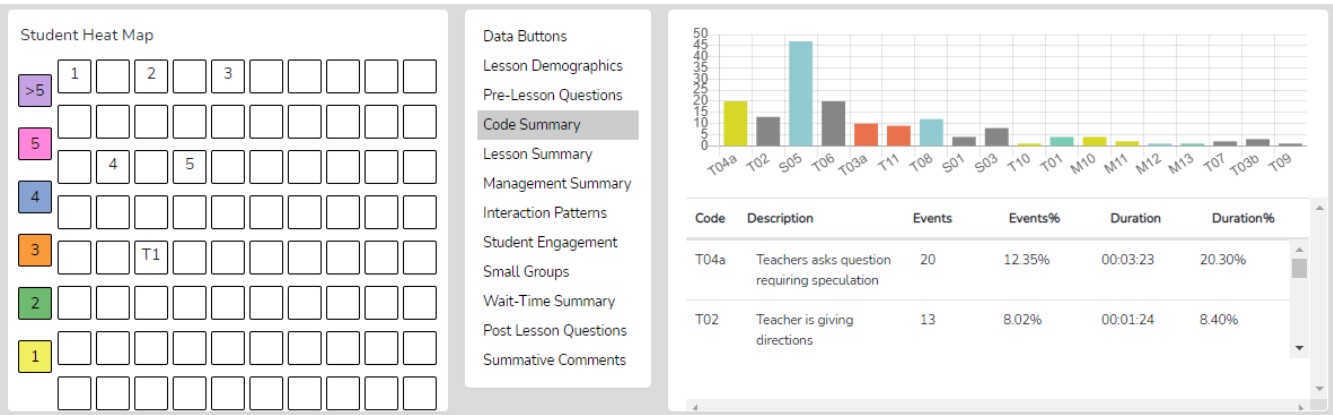

**Figure 2.** Code summary example. **Note: The raw counts, time accumulations, and percentages of various discrete teacher and student actions coded during the observation create an evidence-based picture of teacher tendencies that becomes the basis for a more meaningful analysis shown below. However, even the raw data can be an important indicator.** For example, suppose the teacher asked 85 questions during an observation, and 81 were low-level yes/no questions. In that case, it creates a baseline indicator to measure potential or desired change in future classroom observations. Perhaps the next lesson's change in teaching target was to ask a greater number of higher-level questions, and data from the observation shows the teacher asked 35 (out of the 75) questions requiring higher-level thinking. This increase is a distinct change in questioning that informs the teacher on how the change impacted student thinking, an important indicator to monitor regarding growth in teaching skills. As such, even raw counts are valuable indicators and provide a data-enabled window into a teacher's tendencies and an indicator of whether that teacher has changed their approach in a manner supported by research (see Figure 3).

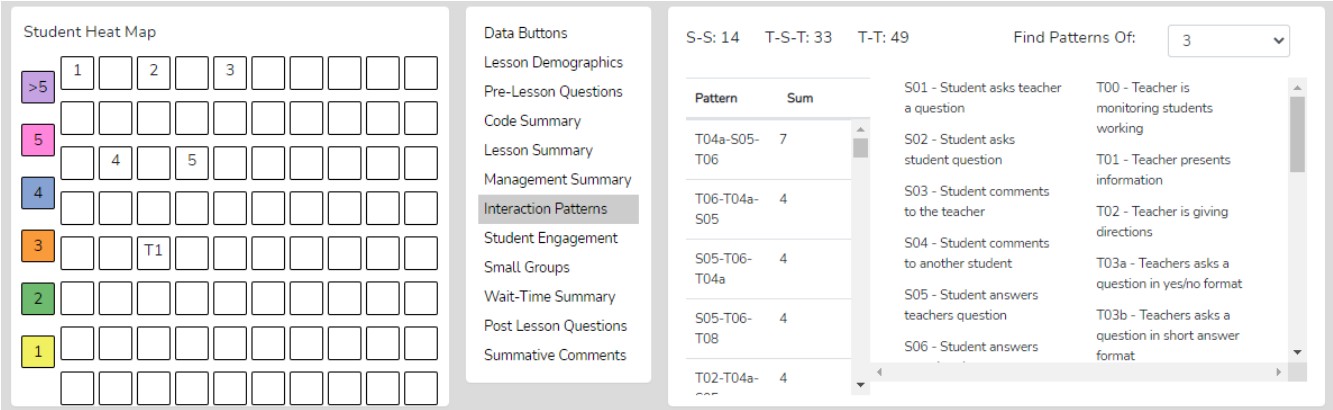

**Figure 3.** Teacher–student interactions patterns example. **Note: A teacher's pattern of interacting with students in the classroom can be identified immediately following the observation.** Altering this interaction pattern even with subtle changes can profoundly affect the learner and learning. Also, as noted in Figure 3, the indicator of how many times students interacted with each other in this lesson is fourteen (S-S 14). These data arise from teacher actions contrary to what is often recorded in the typical teacher-centered questioning and responding interaction pattern. When a teacher changes some behaviors when interacting with students, this student engagement factor (S-S) and number can be increased, with resulting benefits for the teacher and learner (see Figure 4).

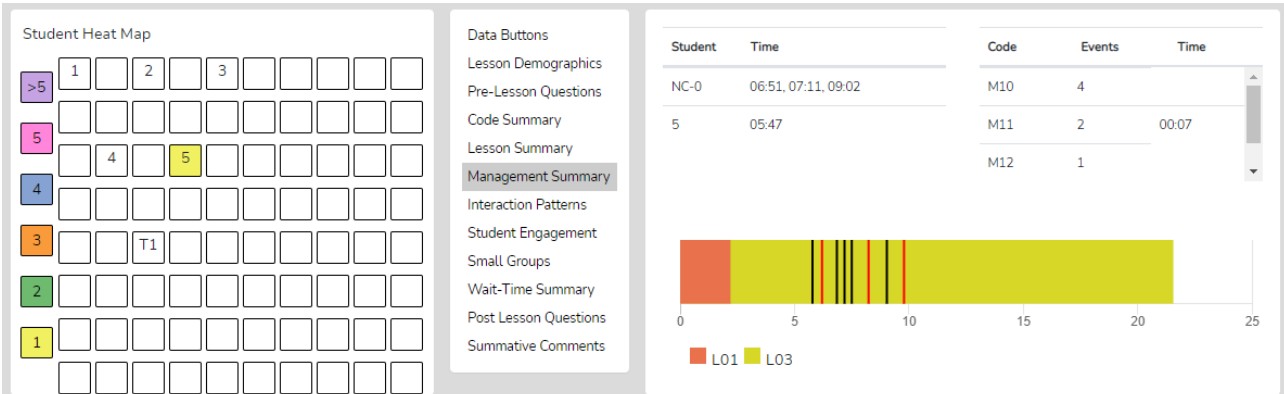

**Figure 4.** Management summary example from an avatar classroom. **Note: The seating chart heat map shows where and how many behavioral issues were observed. As seen in Figure 4,** on the timeline, the black bars are markers that indicate when in the timeline of the lesson, the behavior issues occurred; markers are also linked to the video of the misbehavior. Teacher interventions are indicated by red bars and linked to the lesson and video. For more detail, misbehavior data can be parsed by individual student behavior. For example, perhaps a student is an attention seeker, and the teacher is not adept at handling this type of misbehavior. If data are collected using the seating chart, the observer and teacher can single out data for that student to show where and when on the timeline those behaviors occurred and then examine the video to see how those misbehavior interactions were managed(see Figure 5).

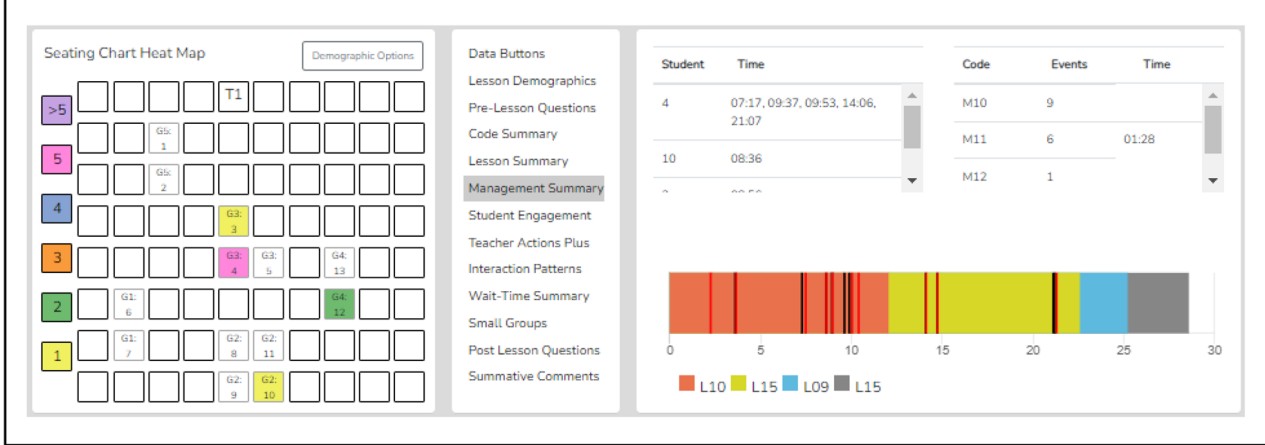

**Figure 5.** Management summary example from a regular classroom. Note: Two students each exhibited one misbehavior, one student exhibited two misbehaviors, and one student exhibited five misbehaviors. Figure 5 shows that it is visually obvious that misbehaviors occurred toward the area of the classroom that is farthest away from the teacher, which often happens when teachers are anchored in the front of the room. The data can help a teacher identify more specific targets for classroom management interventions and document if interventions have the desired effect (see Figure 6).

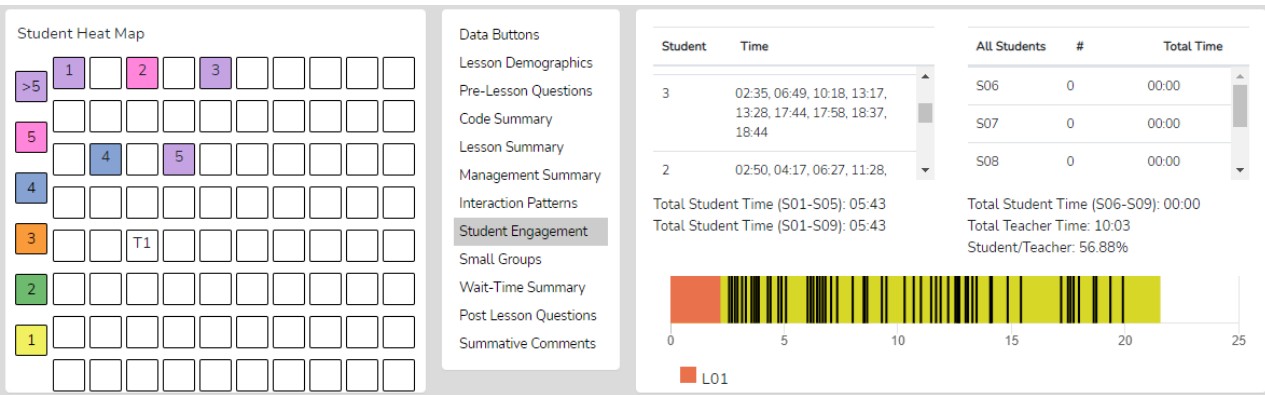

**Figure 6.** Student engagement summary example from an avatar classroom. **Note: The seating chart provides a visual of student engagement by displaying which students were asking or answering questions and a color-coded level of engagement for each student.** Shown in Figure 6, the black bars on the timeline represent where individual student engagement occurred (also with links to the video) (see Figure 7).

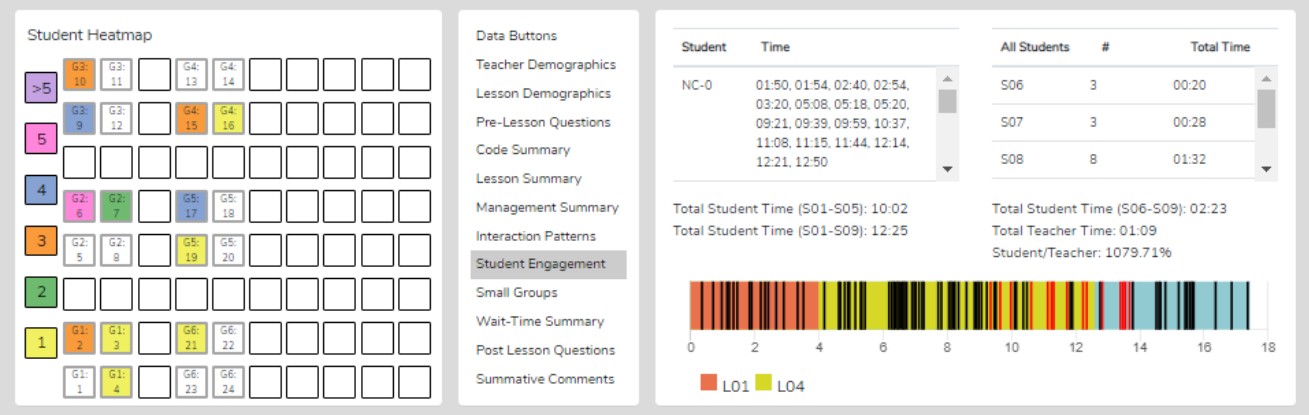

**Figure 7.** Student engagement example from a regular classroom. **Note: Student engagement, as noted in the seating chart heat map, was distributed across many different small groups, and half of the members of the small groups contributed something during the teacher–student interaction phase of the lesson.** In addition, as seen in Figure 7, other data indicated that most individual students were engaged and contributing during the small group work phase within the small groups, so there was a high level of student engagement in the lesson. Note on the lesson timeline that the red bars show when in the lesson, the whole group of students was engaged (also with links to a video). Whole-group engagement might include a think–pair–share or digital device response or another activity in which all students interacted or responded in some manner.

Other studied potential uses for pre-service or in-service teacher development include the following: (1) working with students with special needs (for example, Vince Garland and colleagues [24] worked with and studied the effects of evidence-based practice on students with autism and found an increase in implementing discrete trial teaching from 37% to 87% after treatment of six 15 min sessions. Additional studies demonstrating impact include (1) special education (Whitten and colleagues [25], Rees and colleagues [26], Anzelmo-Skelton and Ratcliff [27], and Walker and colleagues [28]); (2) developing classroom management skills, which include thwarting bullying behaviors (Lassman and colleagues [29], Floyd and colleagues [30], and Ludlow [31]); (3) developing skills necessary to facilitate inquiry science (Sander [32]); (4) teaching accommodations for English Language Learners (Regalla and colleagues) [33]; and (5) developing parent–teacher conferencing skills (Powell and colleagues [34] and Walker [35]). This vast array of research combined

with data collection in simulation and practice has an array of applications to science teacher education.

### 3.1. An Authentic Experience

Why does teaching in front of the avatars seem real for novice and experienced teachers? A primary reason that the interactions between the teacher and the avatars seem more realistic than a game is that the avatars consist of both computer- and human-controlled parts of a patented process created at UCF. In a review of the literature, Hayes and colleagues [36] found that teachers reported high levels of presence and reality in the simulated environment. The source of the voices is a real person. As such, the questions posed by the avatars to the teacher or the avatar's answers given to questions posed by the teacher are responses generated through improvisation shaped by the content expert (in this case, the science teacher educator) exhibited by the interactor, also known as a puppeteer, in the context of the current lesson. Like typical teenagers, the avatars also know about current music and other tidbits about current events, sports, pop culture, and books that teenagers might read. The role of the trained interactor is to stay current on content and to know the context of the environment (e.g., weather in Hawaii versus Alaska) and local sports, music, and demographics to create as real of a context as possible. Again, the point of the simulator is not to replace practice but, like those simulators used in the military, in medicine, and in aviation, to simulate aspects of a targeted skill to be learned or remediated in a low-risk environment and with the potential for repeated practice until mastery [37].

Simulation settings provide a less complex environment for practice [38]. Many novices working to develop teaching skills in a classroom of 24, 30, or even 45 students can become overwhelmed with such environments. However, a simulator featuring only a few students allows a safe chance to practice. The impediment to effective instruction often labeled as number one by novices (and many experienced teachers) is the challenge posed by inappropriate student behavior. Classroom management, proactive or reactive, is a troublesome central factor to overcome and get beyond when teaching a lesson. Inappropriate behavior by a few students can derail instruction and lead to burnout and stress [39]. One of the goals of a simulated classroom is for future teachers to practice teaching skills without putting classroom students at risk [40,41]. The avatars mimic reality but can also display raised or lowered levels of inappropriate behaviors, such as an attention-seeking student who is quite happy to respond ad nauseam, or more difficult challenges, such as aggressive and revenge-type behaviors, or passive fear of failure behaviors. Fortunately, levels of behavior can be determined ahead of the session by the subject matter expert, giving novices very minimal challenges or elevating them to the level best suited for their current skill level or the level of challenge that they might see in the context of the classroom environments in which they will be placed.

Finally, this simulation setting allows for ideal videotaping, reflection, and analysis possibilities, allowing for "virtual rehearsal" to ensure practice over and over again to the level of the participant or the teacher educator's satisfaction. For researchers, the best part of these avatar students is that they have no "real" parents and, therefore, no real concerns about videotaping minors or human subject approval creating a perfect setting for randomized studies. Simple video technology can easily capture the teaching session with both the avatars and teacher included in the video at all times, with high-quality audio that negates the poor audio issues often resulting when capturing video in a real classroom.

### 3.2. Specific Use of TeachLivE

In the authors' institutions, one of the uses of TLE TeachLivE has been in pre-service science teacher preparation. Undergraduate science majors and minors can opt for a taste of teacher education by enrolling in the course Introduction to Teaching Science and Mathematics. In this course, students utilize TLE TeachLivE twice, teaching 10–15 min lessons. Once admitted into the middle and secondary science teaching program, students

teach two to four additional 10–15 min lessons. Participants plan a short lesson and then teach to the avatar students, who provide realistic content and behavioral responses. The purpose of these lessons is two-fold: (1) to practice and refine their questioning, responding, and use of wait-time interaction skills, and (2) to measure their ability to recognize each of the four types of student misbehaviors and the intervention and response that are appropriate for de-escalating behaviors, versus responses that serve only to escalate and inflame the situation [42,43].

The session is recorded, capturing both the teacher and the avatars, with the video uploaded for viewing, analysis, reflection, and coaching using the SeeMeTeach observation tool (described in the next section). These lessons are processed with the pre-service teachers in a manner similar to what Baird and colleagues [44] describe as structured reflections and what TLE TeachLivE researchers label the after-action review process [36].

Using traditional methods, the instructor can provide post-teaching feedback, and peer groups who observe can discuss the teaching experience and reflect on the experience. But what is possible using an additional data extraction tool with these observations?

## 4. An Evidence-Based Teacher Observation Tool

### 4.1. Evidence Versus Qualitative Impressions

Preparing teachers to teach effectively and engage learners at high levels involves complex tasks that raise knowledge and awareness through practice, observation, and data collection, followed by analysis and reflection. An iterative cycle of observation and reflection is essential for changing teaching practice [45]. Embedded into the complex act of teaching are numerous teacher and student actions and responses, occurring in a short amount of time, so much so that novices are left with general impressions and memories of their actions during the teaching episode that often miss the mark.

As such, novices depend highly on observer feedback, much of which is qualitative [46]. For example, written comments are a common aspect of observation and are mostly qualitative. The authors are not trying to dismiss or reduce the potential value of an observer's comments, yet comments often lack reference to any actual data. Likert-scaled feedback forms are largely qualitative impressions, as are rubrics which often include several factors in a single column using a ranking scheme. The typical teacher observation captures or includes very little data.

However, most professions embrace and utilize some quantitative indicators of impact or progress. Why not teacher observation? We contend that during the observation and feedback process, pre-service teachers and teacher educators should use more quantitative indicators of teaching. Whether during the teacher development phase or teacher assessment process, feedback and coaching should and now easily can use more evidence-based indicators. While the authors recognize the value of qualitative comments and feedback (which is why the tool also contains a qualitative mode), what has been missing from teacher observation is feedback based on the immense data mined from a teaching episode, the driving force for the development of the teacher observation tool.

What data can one glean from a teaching episode to help a teacher improve their actions in a manner consistent with maximizing student engagement and learning? Teaching episodes can generate an immense amount of data related to what teachers do, such as questioning, responding, and use of wait-time, while generating large amounts of data about student engagement or misbehaviors. An observer might collect upwards of 240 data points during a 20 min lesson. Specific data could potentially be collected during a teaching session (from video or real-time) inclusive of (1) types of questions and responses, (2) average and specific wait-times, (3) specific types and lengths of teaching strategies utilized, (4) specific types of interchange between students or general student participation, (5) predominate patterns of interactions between the teacher and students, (6) student engagement data specific to individual students and also delineated by demographics such as gender, learning disability, minority status, or ELL/Bil., (7) misbehavior data at the individual student level, and (8) teacher intervention in response to misbehaviors

(individual or whole group). These types of factors framed the development of the teacher observation data collection tool.

This tool addresses an age-old problem of gathering a subset of the potential data without using more than one observer and without experiencing cognitive overload. Observers using typical pen and paper or laptops are limited concerning what they can attend to, collect, and compile. Hence, a running commentary with post-lesson qualitative feedback forms has been the norm. But what is possible if technology is employed, much as has occurred with simulation and TeachLivE? Research indicates that when teacher candidates are provided opportunities to reflect upon and discuss classroom practices, their understanding of the teaching situation deepens [47]. What are the possibilities when data are added to the observation and feedback cycle?

Pilot efforts showed success with using teacher observation software to gather more quantitative data during observation [48–50]. Using the tool demonstrated that data collected during a teaching session could include all of the above factors without the observer reaching cognitive overload.

*4.2. Current State of Data-Gathering and Analysis Tool*

This piloted teacher observation tool is now an online web-based app called SeeMe-Teach (SMT) and is device-agnostic. This tool emerged as a part of the work of observing teachers in a simulated classroom but has evolved into a stand-alone application. First, the readers will learn about the observation tool's extensive data-gathering and analysis components. This is followed by a presentation of data that highlights using the power of technology for simulated classrooms and enhanced observations to enrich teacher preparation and refine key practices in science education.

As this tool was developed and used to elevate the feedback–teaching cycle and future teachers' skills, two components seemed essential: (1) having access to a video of their teaching as "proof" that legitimizes data or an observer's comments, feedback, and coaching suggestions, and (2) receiving feedback and coaching based on data collected about their teaching either by itself or in conjunction with qualitative commentary to lend credence to an observer's impressions. As such, the current state of the evidence-based teacher observation tool is as follows:

- This tool can be used either in real-time or with a synced video.
- Observers can view the video while linking time-coded commentary to the lesson with category headings for a qualitative-type observation.
- Observers capture numerous data points that become evidence-based indicators of the teacher's performance and skill level.
- Data include teacher actions and decisions, including but not limited to questioning, responding, and use of wait-time; student actions related to student engagement and misbehaviors; and the type of lesson in play, such as lecture, reading groups, or small-group work.
- Data analyses are instantaneous upon completion of the observation, with critical factors displayed in various visual representations including graphs, charts, tables, and heat maps generated.
- If a video of a teacher teaching a lesson is used for data collection, all data points are linked to specific video segments for use during the feedback phase so that coaches can point out examples of practice and novices can see themselves in action. Video showing the example is much more powerful than a verbal description of the same event and reduces a novice's tendency to discount commentary from the observer.
- The observer can now use the data to form the basis for feedback, discussions, and coaching toward improving teaching.

Data instantly analyzed upon completion of the observation provides opportunities to identify and target specific components of teacher–student discourse and provide feedback and coaching. When using the SMT tool, what is now possible, and what questions can be

asked and answered based on collecting data that become evidence-based indicators for feedback and coaching? This feedback can include the following:

- A complete profile of all teacher actions and teacher–student interactions in the lesson to show the predominant behaviors and teacher tendencies.
  - What types of questions were asked by the teacher, and how many of each type?
  - What types of teacher responses followed student actions, and how many of each type?
  - What were the wait-time averages and specific wait-times for each teacher and student action?
  - How did the above fit with the targeted goals of the lesson and the manner of interaction the future teacher suggested that they would enact in the lesson?

- A complete profile of all student actions, showing interactions with the teacher and other students and student misbehavior.
  - Which students are interacting, and which are passive?
  - Are most questions answered by a few students, while the other students are satisfied to be non-responsive throughout the lesson?
  - How did the teacher employ strategies that engaged most or all students?
  - Were students with special needs or ELL students engaged at a level comparable to regular education students?

- An analysis of the data uncovering the critical patterns of teacher–student interactions.
  - When teachers ask questions and students respond, is it a productive pattern or one contrary to the lesson's goals?
  - If student engagement and thinking is the goal, are open-ended questions present or absent, or were all follow-ups to student responses the teacher clarifying instead of asking the student to explain their answer further?

- An analysis of small group member interactions and teacher–student interactions.
  - Small groups are often semi-productive, with a subset of members doing most of the work. What did the data indicate about equity among small group members regarding work and product generation?
  - What was the nature of the teacher's interactions with the small group, and did the interaction support or did instruction undermine the goals of the lesson?

- A complete profile of student misbehaviors and how the teacher handled such behavior.
  - Are misbehaviors initiated by a few students versus many? Where in the classroom are the misbehaviors occurring? What is the extent of misbehaviors without teacher intervention?
  - What can be learned if misbehavior counts are high during x type of lesson and low during y type of lesson?

For greater insight into the data analysis mode, the following are screenshots taken from observation of an avatar classroom (and a regular classroom to show how this translates to both a technology-enhanced and "real" classroom setting). These figures provide the reader with examples of the data analysis mode of the teacher observation tool. The comments below the figure are a sample of what was gathered during this observation and do not represent the totality of data and analysis an observer can glean and use for feedback.

The discussion of the SMT was intended to show the many varied data points and features of the technologically enhanced observation tool and how data-gathering and evidence use might be an extensive component of teacher observation. Combining the SMT with the simulated classroom, TLE, is discussed to show the power of and synergy between these tools for teacher performance and, most importantly, teacher learning in an initial pilot study.

**5. Merging Use of TLE TeachLivE and SeeMeTeach: Do Teaching Fingerprints Emerge?**

Teaching practices differ and have distinguishing characteristics. Changes to teaching practices can result in subtle or major effects on the learning environment, learner, and teacher. Berg [9] suggests that all teachers have a predominant manner of teaching that is identifiable, much like a fingerprint. Quantitatively, teachers have tendencies that can form patterns of instruction that have a major impact on student thinking, learning, or engagement in instruction. These patterns of interaction can be consistent from one lesson to the next and might be as identifiable as a fingerprint. Fingerprints form from how teachers ask questions, respond to student answers, and use wait-time. These are core aspects of the teacher–student interaction and are a major factor in teachers being the most important school-based factor influencing student learning and academic achievement.

Why is collecting data and profiling how teachers interact with students so important, especially during the development of future teachers? Identifying these tendencies and patterns is central and a first step when altering discourse between teacher and student, which can elevate or undermine a lesson or an activity. Concerning developing outstanding teachers, even subtle changes in how teachers interact with students can profoundly affect the learner, and learning and teaching fingerprints can be modified. Teachers should have the skills to alter their interaction fingerprint to support the specific goals of instruction for that lesson and intentionally exhibit a consistent core of interaction skills that supports the targeted goals for that classroom as a whole.

The authors suggest and, as noted previously, research has shown that the TLE TeachLivE simulation demonstrates a realistic environment in which to practice and, therefore, potentially improve one's teaching skills. Teachers in front of the avatars exhibit behaviors that form an identity, like fingerprint evidence, resulting in their teaching profile. This evidence serves as a valid indicator of their current teaching skills. In short, data collected using SMT show that simulation episodes offer possibilities for diverse teaching approaches and allow for variation and differences between teachers to emerge—fingerprints become visible even in short episodes when teaching avatars. A case in point is the data collected using SMT from two teachers in TLE TeachLivE sessions, which provide both tendencies and a fingerprint of each teacher and uncover the variation between the two.

Table 1 contains compiled raw data of T (teacher actions) and S (student actions) codes noted during an observation, reported by counts of each event by accumulated time and percentages of each. The notable differences between Teachers A and B are highlighted in bold print. First, note that when comparing Teacher A vs. Teacher B for overall T and S codes, Teacher A's percentage of the total time was 67% for T codes and 33% for S codes, which means the teacher was talking about two-thirds of the time. Contrast those numbers to Teacher B, who exhibited just the opposite, where T codes amounted to 30%, and S codes were 70% of the total time, meaning that students responded much more than the teacher was talking. This ratio is more consistent with indicators of a student-centered classroom where students are highly engaged [51]. Analyzing the various counts and percentages of the S (student) and T (teacher) codes allows researchers, observers, or practitioners to determine and document the teacher's tendencies to incorporate various behaviors into their teaching that foster student engagement in their lessons. In Table 1, the data in bold print highlight important and contrasting differences between Teachers A and B. Such data are critical for feedback, deconstructing the teaching episode, and emphasizing how data should guide setting targets for growth and change in teacher practices.

The teacher actions in Table 2 break down the interactions into more discrete events. Teacher A and Teacher B's respective five most utilized codes in terms of total time are shown in Table 2. Teacher A has more teacher-centered tendencies, consisting of asking short-answer questions, providing information, giving directions, and answering student questions, and 22% of the time, students answer the short-answer questions.

**Table 1.** Raw data collected—Teacher A vs. Teacher B.

| Specific Code and Action | Teacher A | | | | Teacher B | | | |
| --- | --- | --- | --- | --- | --- | --- | --- | --- |
| | #of Events | % | Total Time | % | #of Events | % | Total Time | % |
| S1—Student asks T a question | 19 | 6.8 | 76.6 | **6.5** | 11 | 6.0 | 42 | **2.4** |
| S3—Student comments to the T | 12 | 4.3 | 50 | **4.3** | 8 | 4.3 | 23 | **1.3** |
| S4—Student comments to another S | 2 | 0.7 | 8.3 | 0.7 | 0 | 0 | 0 | 0 |
| S5—Student answers T question | 70 | 25 | 261 | **22** | 47 | 25 | 1176 | **66** |
| T1—Teacher presents information | 22 | 7.9 | 180 | **15** | 15 | 8.1 | 142 | **8** |
| T2—Teacher is giving directions | 7 | 2.5 | 110 | **9.3** | **0** | **0** | **0** | **0** |
| T3a—Teacher asks yes/no question | 22 | 7.9 | 49 | 4 | 33 | 18 | 90 | 5 |
| T3b—Teacher asks short-answer question | 32 | 11 | 195 | **16** | 7 | 3.8 | 48 | **2.7** |
| T3c—Teacher asks question—speculation | **0** | 0 | 0 | 0 | **10** | 5.4 | 48 | 2.7 |
| T4—Teacher asks question—speculation and justification | **0** | 0 | 0 | 0 | **2** | 1 | 10.7 | 0.6 |
| T5—Teacher rejects student answer | 2 | 0.7 | 7.6 | 0.64 | 2 | 1.1 | 6.7 | 0.4 |
| T6—Teacher acknowledges S answer w/o judging | 10 | 3.4 | 15 | 1.13 | 12 | 6.5 | 24 | 1.3 |
| T7—Teacher confirms student answer | **30** | 11 | 40 | **3.4** | **8** | 4.3 | 10.5 | **0.6** |
| T8—Teacher repeats student answer | 26 | 9.3 | 47.8 | 4 | 11 | 6 | 38.7 | 2.2 |
| T9—Teacher clarifies the answer for the student | 3 | 1.1 | 8.9 | 0.7 | 2 | 1.1 | 3.2 | 0.2 |
| T10—Teacher answers the student's question | 19 | 6.8 | 125 | **11** | 8 | 4.3 | 63.7 | **3.6** |
| T11—Teacher asks the S to clarify their answer | 3 | 1.1 | 8.9 | 0.7 | 2 | 1.1 | 3.2 | 2 |
| Overall T Codes | 176 | 63 | 788 | **67** | 119 | 64 | 540 | **30** |
| Overall S Codes | 103 | 37 | 397 | **33** | 66 | 36 | 1242 | **70** |

**Table 2.** Contrasting Tendencies Between Teacher A and Teacher B.

| Teacher A | Total Time % | Total Time % | Teacher B |
| --- | --- | --- | --- |
| S5—Student answers teacher question | 22 | 66 | S5—Student answers teacher question |
| T3b—Teacher asks short-answer question | 16 | 3.7 | T10—Teacher answers student question |
| T1—Teacher presenting information | 15 | 2.7 | T3b—Teacher asks short-answer question |
| T10—Teacher answering student question | 11 | 2.7 | T3c—Teachers asks question requiring speculation |
| T2—Teacher is giving directions | 9 | 2.4 | S1—Student asks teacher a question |

Conversely, three times as much of Teacher B's lesson was spent with students answering questions. Teacher B asked some questions requiring speculation (compared to none for Teacher A), and speculation questions often resulted in longer student responses. Teacher A also presents much more information than Teacher B. Regarding comparative observation of both teachers and classes, the qualitative impressions would have been different. However, the underlying data explain the qualitative differences, as the data collected specifically point to particular tendencies and instruction patterns exhibited by each teacher. These are the targets that provide the roadmap for change if teaching is to improve.

From data collected during a TLE TeachLivE session, the SeeMeTeach observation tool also generates an interaction pattern analysis, displaying a hierarchy of predominate 3, 4, 5, and 6 code patterns that provide evidence for the user to conclude whether patterns of discourse between themselves and their students are congruent with the goal(s) of instruction, or not. These pre-service teachers were instructed to teach a lesson to find out what students knew about a topic. Tables 3 and 4 provide Teacher A's and Teacher B's instruction patterns. Patterns indicating attempts to uncover a deep understanding of what students knew were not evidenced to much degree except for Teacher B, who exhibited a T3c-S5-T6 pattern four times (teacher asks a question requiring speculation, a student responds, the teacher acknowledges without judging). Note that this pattern that asks students to answer a more open-ended question requiring speculation, which is a pattern more indicative of inquiry and a pattern leading to higher levels of student thinking, is not intuitively used and often needs to be taught to future teachers. Then, Teacher B follows

by acknowledging without judging, which often leads to more student thinking and more responses by the student. If a four-code sequence was reported and the fourth code were T11 (teacher asking the student to clarify), there would be evidence that the teacher was probing for more information from the student and asking the student to clarify instead of what teachers normally do, which is clarifying for the student—a teacher behavior that shuts down student thinking and lessens the teacher's understanding of what students know and understand.

**Table 3.** Three code discourse patterns for Teacher A.

| Count | Code |
|---|---|
| *11* | T3b-S5-T8 = teacher asks short-answer question, the student responds, the teacher repeats student's answer |
| *9* | S5-T7-S5 = student responds, teacher confirms, the student responds |
| 9 | T7-T3b-T7 = teacher confirms student response, teacher asks a short-answer question, teacher confirms student response |
| 9 | S5-T8-T7 = student answer, teacher repeats student answer, teacher judges answer |
| 7 | S5-T7-T3b = student answer, teacher judges answer, teacher asks short-answer question |

**Table 4.** Three code discourse patterns for Teacher B.

| Count | Code |
|---|---|
| 10 | S5-T3a-S5 = student responds, the teacher asks yes/no question, the student responds |
| 10 | T3a-S5-T3a = teacher asks yes/no question, student responds, teacher asks yes/no question |
| 5 | T3a-S5-T8 = teacher asks a yes/no question, student responds, the teacher repeats student answer |
| 4 | T3c-S5-T6 = teacher asks question requiring speculation, student responds, teacher acknowledges without judging |
| 4 | S1-T10-T3a = student asks teacher a question, teacher responds, teacher asks a yes/no question |

To summarize the importance of collecting data on teacher actions, student actions, and other occurrences in teacher practice, the SMT tool can collect raw counts that define teacher tendencies and examine the sequence of codes to determine the most to least common interaction patterns in the teacher–student discourse. These patterns serve as fingerprints or identifiers of the teacher's current teaching skills and perhaps guide what changes are needed regarding teacher–student interactions during instruction. The research-based knowledge of effective instruction [13] suggests that not all patterns produce the same outcome; a pattern used for direct instruction would not generate the positive effects hoped for when teaching an inquiry lesson. Teacher patterns and tendencies have often been part of the often-untapped data available to an observer. Shymansky and Penick [50] demonstrated long ago how gathering data on teacher actions can help guide restructuring of learning environments that will result in noticeable differences in learning outcomes, such as levels of student engagement and increased interaction between students. Observers and teachers now have access to a plethora of data. Such data and real-time analysis, afford a new lens to view teaching, allowing users to uncover and view the discrete details and instruction patterns with clarity.

## 6. Conclusions

The researchers examined the following questions: (1) How can the teacher observation tool help identify and collect data during a teacher observation regarding key and discrete factors of teacher–student interactions and student engagement when attempting to improve teaching effectiveness? And (2) Does the TLE TeachLivE simulation produce a realism that offers potential for a wide enough variation in the display of teaching skills so that teaching fingerprints emerge?

First, the SMT tool was developed to collect objective data to be used by an observer in teacher education or schools to help a future or practicing teacher improve his or her teaching practice or used independently by a teacher for data collection and self-reflection. SMT provides numerous indicators and analyses of teacher and student actions and engagement in the lesson. The observation software allows the user to collect and analyze data while providing many different visual representations, including tables,

graphs, and heat maps of seating charts to use in the feedback process. Most data points are linked to the video for instant viewing, another valuable aspect in the feedback and coaching session when examining teaching for key indicators.

This study has shown that an evidence-based observation tool can collect a substantial amount of quantitative data while observing a teacher who is teaching while immersed in a simulation. With instant analysis, the observation tool produces indicators and critical factors available for feedback, coaching, and data-anchored self-reflection. In addition, since many observers are mostly grounded in and attuned to the qualitative aspects of observation and feedback, using the technology and engaging in quantitative data collection with related feedback is similar to having a new lens from which to view instruction. Access to these data can benefit both teacher and observer regarding improving teaching skills.

Second, the TLE TeachLivE simulation is a prime and low risk setting to practice questioning and responding skills. SMT-collected data provide explicit evidence of tendencies and patterns of interaction during teaching simulation episodes. In this study, the data indicated that future teachers were ripe for improvement in teaching skills regarding teacher–student discourse in terms of questioning and responding to get students thinking more deeply about the content and more engaged in the lesson. In short, these indicators are critical and timely feedback to the teacher on whether their teaching practices match their instructional goals or whether teaching practices might be altered when entering the next round of the simulation.

Third, since this simulation offers a more controlled and low risk setting than a classroom containing real students, it is a teacher preparation tool suitable for practicing teaching skills and assessing a pre-service teacher's ability to exhibit particular skills targeted by teacher education programs. It would appear that TLE TeachLivE, as a virtual avatar classroom, is realistic enough that teachers can demonstrate different approaches to instruction, and even a fifteen-minute lesson taught to avatars produces contrasting data between individual teachers. Data collection using SeeMeTeach not only gathers baseline data but allows participants to identify and recognize changes in teaching due to the use of interventions such as TeachLivE. As such, it has the potential to become a vital companion tool for users of TeachLivE.

Finally, there is synergy at work when a simulation for practice teaching is combined with an evidence-based teacher observation tool in an attempt to change and measure the change in teacher actions or student actions in a simulated practice teaching setting. The simulation offers opportunities to practice teaching and practice observing in a low-risk setting. The teacher observation tool elevates the observer's capabilities to use data as evidence-based feedback that goes far beyond the observer's impressions. Such data and evidence help teachers target what to modify in the next simulation round and note with confidence whether teaching practices have changed or not. Those who prepare teachers and those preparing to teach stand to benefit from low-risk simulation practice teaching technology and from using a tool purposed for gathering data as indicators of teaching and then using those data as the core of evidence-based feedback and coaching.

**Author Contributions:** All authors contributed equally to all aspects of the published version of the manuscript. All authors have read and agreed to the published version of the manuscript.

**Funding:** This research received no external funding.

**Institutional Review Board Statement:** The study was conducted in accordance with the Declaration of Helsinki, and approved by the Institutional Review Board (or Ethics Committee) of University of Wisconsin-Milwaukee, IRB 16.168, December 2015, and IRB 21.174, January 2021.

**Informed Consent Statement:** Informed consent was obtained from all subjects involved in the study.

**Data Availability Statement:** The data presented in this study are available on request from the corresponding author.

**Conflicts of Interest:** The authors declare no conflict of interest.

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
