# Peer review of "Using a Virtual Avatar Teaching Simulation and an Evidence-Based Teacher Observation Tool: A Synergistic Combination for Teacher Preparation"

_education, doi:10.3390/educsci13070744_

Round 1

Reviewer 1 Report

The introduction section of the manuscript is identical to the abstract.

The claims made about teaching skills in lines 55-62 are not supported by recent citations.

The presentation of the TLE TeachLivE simulation in lines 95-112 is interesting but seems heavily biased by the authors' use of the program. Instead, a more neutral position should be used to introduce the program to the reader. Additionally, the explanation of the authenticity of the program in lines 148-192 makes a number of claims that are unsupported by citations from the literature. Because of the lack of supporting literature, this section reads more like a sales pitch for the program than a balanced presentation of information.  

Although the presentation of the SMT tool appears to carry a great degree of bias, I applaud the authors for their thorough description of a tool to gather quantitative teaching data and how that data can address common issues in teacher preparation research. 

The presentation of the raw data in lines 472-566 is interesting but is more of an example of the data collection previously described than research findings. Additionally, the formatting of these tables makes interpretation difficult. The inclusion of the tables at the end of the document is helpful, but reformatting for those included in the manuscript is necessary. 

Throughout the manuscript, the authors frequently switch between the third person (the authors) and first person (our/we). Either is acceptable, but both should not be used in the same document.

Citations are missing or have incorrect years of publication between the in-text citations and the references list. A thorough check for congruency is needed. 

Overall, I find the data collection methods presented in this manuscript interesting. However, the purpose of the research is unclear. Parts of the document seem to be an attempt to validate data collection methods but lack sufficient rigor in testing to adequately do so. Other parts of the document are written like a case study of the use of the technology but lack much-needed depth from the practitioner's viewpoint. I believe that the use of the technology described in this manuscript is valuable and would be a very interesting conference presentation or popular publication dedicated to practice. However, significant revisions to this work, particularly in style and research methodology are required to create a validation-based methodology study or an evaluation-based case study.

The English language quality in the manuscript is very good, although the writing is often unnecessarily wordy (e.g., lines 132-146). 

Author Response

Perhaps more references could be made to the last three years.

** Added new references from recent years.

Reviewer 2 Report

Thank you for choosing our journal to share your interesting and relevant research data. The research is scientifically and practically sound. The research questions are clearly and coherently justified and argued, there are sufficient details given to replicate the proposed research procedures and analysis. The interpretation of the data presented is thorough. Perhaps more reference could be made to publications from the last three years. 

The research has shown that the TLE TeachLivE simulation demonstrates a realistic environment in which teachers in front of the avatars exhibit behaviors that form an identity, SMT-collected data provides explicit evidence about trends and patterns of interaction during teaching simulation episodes.

The article is well-written and easy to understand. The paper can be accepted without any further changes.

Author Response

Reviewer 2

References relevant to the research and adequately referenced

Improved with added references

The design and intent of this work and results need to be clearly presented

Edited accordingly in two sections of the document

Claims made about teaching skills in lines 55-82 are not supported by recent citations.

Added recent citations

Lines 95-112 seem biased

Edited to be more neutral

Lines 148-192 make claims unsupported by the literature

Added supporting citations

The presentation of SMT tool appears to carry a great deal of bias

Tried to tone down, but describing the many data gathering and analysis capabilities may sound like a promotion, but it is not.

Authors switch from 1st to 3rd person

Edited the manuscript to be consistent with 3rd.

Missing citations – congruency check is needed

Done

Unnecessarily wordy in lines 132 – 146

Edited to two sentences.

Table 1

The reader needs to know the extent of data gathering and analysis potential and how some data points are substantive instruction identifiers. The authors bolded the contrasting data points between teacher 1 and teacher 2 so the reader could easily identify key data points. It is well worth the one page to keep the whole of Table 1.

Tables 2, 3 and 4

Maintained these Tables as they are succinct.

Round 2

Reviewer 1 Report

Thank you for addressing most of the concerns presented in the previous review.